# Utilizing Simulation Software to Develop Injection Molding Process Windows with High-Impact Polystyrene

**DOI:** 10.3390/polym17060718

**Published:** 2025-03-08

**Authors:** Travis O’Leary, Rachmat Mulyana, Jose M. Castro

**Affiliations:** Industrial and Systems Engineering, College of Engineering, Columbus Campus, The Ohio State University, Columbus, OH 43210, USA; oleary.161@osu.edu (T.O.); mulyana.1@osu.edu (R.M.)

**Keywords:** injection molding, simulation, process window, controllable process variables

## Abstract

While design groups have utilized the abilities of injection molding simulation software, its use is often underutilized by process engineers. To expand the application of simulation software to manufacturing groups, this work focuses on developing a methodology to construct injection molding process windows through the predictions of simulation software. The methodology was developed by testing combinations of controllable process variables in the filling and packing stage of the injection molding process with high-impact polystyrene. Using this method, the process window can be tailored to a manufacturer’s desired product performance measures as well as target a specific defect they are facing. The process windows developed were experimentally validated displaying a successful combination of controllable process variables in the filling and packing stages which resulted in an acceptable part. Additionally, the process window was able to predict the dimensional shrinkage of a part within 1% of the experimentally produced part. This analysis establishes confidence in the software’s ability to aid manufacturing groups to successfully run their operations.

## 1. Introduction

Injection molding is a widely used, cost-effective, manufacturing process able to mass produce thermoplastic parts for a wide range of products such as medical equipment, automotive components, and packaging [1,2]. For each stage of the injection molding process, there are several controllable variables that can affect the overall quality of the final product [3]. The most relevant include injection time, melt temperature, mold temperature, injection speed, packing pressure, and cooling time. If one or more of the controllable parameter settings are incorrect, the produced part contains one or more defects [2]. Defects are any undesirable issue with the final product making it unacceptable to the customer. Common defects in injection molding include warpage, flash, weld lines, sinks, and many others. Each defect has a different effect on the final part, and they all result in the loss of time and money [3,4,5,6]. The mechanical properties of products can also be affected by an incorrect process or even after the part has been manufactured, such as if exposed to UV [7]. To ensure the elimination of defects, an operator needs to properly establish the controllable parameter settings. Unfortunately, in most cases, the knowledge required to properly set up a process comes from either trial and error or previous experience [1,8].

The goal of the injection molding industry is to develop and produce a quality product as accurately and quickly as possible. This is achieved by developing a process that will optimize cycle time for a part without incurring any defects. With the multitude of controllable process parameters, many opportunities present themselves to produce defects in a part when the correct combination is not used [9]. Furthermore, as the injection molding industry continues to expand, so does the complexity of custom tooling and part designs. It is because of the development of a process and the designing of a tool that the use of simulation has grown in the industry for both designers and process engineers as they develop cost-effective, quality parts [9].

The aim of the simulation is to select the best combination of process conditions before actual manufacturing occurs through computer-aided engineering (CAE). CAE is a tool that allows the developers of a custom mold to identify issues in its design before construction has begun. Some of the uses include, but are not limited to, how a gate type will affect the flow front of the melt, locations of weld lines in the final product, and the proper location of cooling channels and vents in portions of the mold [9,10]. Before the advancement of simulation and CAE, these issues would not have been noticed until the mold was constructed, and the first parts were produced incorrectly. This led to many molds having to be either adjusted to fit the final part’s criteria or scrapped altogether [11].

Although the use of CAE has benefited the injection molding industry, there are limitations to its accuracy. Limitations of a simulation’s ability to accurately predict experimental phenomena come from the simplification of complex geometries as the software solves equations for fluid flow and heat transfer [12,13]. These equations provide the basis for the predictions of how a product would undergo the filling, solidification process, crystallization phenomena, as well as the formation of microstructures experimentally [14]. All of these factors result in differences between the values obtained through each of the simulation software’s predictions and experimental results of filling, packing, and shrinkage [15]. Even with these limitations, simulation software can be utilized as a guide of what one is to expect during the manufacturing of a product.

Simulations are heavily used in the design and development of a mold, but it is often underutilized in manufacturing by process engineers. Injection molding simulation software, such as Moldflow, can simulate each phase of the injection molding process. Manipulation of controllable process parameters of each injection molding phase within a simulation generates a prediction of a final product that could be produced physically with the same process [16]. Manufacturing engineers now can develop a process window by testing different combinations of parameters to identify and eliminate potential defects while reducing costs incurred from trial and error.

Process windows are a set of operating conditions that allow for the manufacturing of an acceptable injection molded part. Process windows appear as window-shaped areas on a graph containing multiple processing parameters that visually indicate which combination of controllable parameters will produce an acceptable part. If the combination of controllable parameters is chosen within the window, then an acceptable part can be produced. Defects appear when the combination of controllable parameters is chosen outside of the window, producing unacceptable parts that need to be scrapped. It is the goal of the operator to stay within the boundary limits of this window for a successful process.

Previous work in our group on this topic focused on the development of process windows through experimental trials [17]. In this work, several controllable process variables were tested to discover which combinations would result in an acceptable injection molded part based on predetermined performance measures. Although this procedure is successful, manufacturing facilities do not have the production time available to develop process windows in this manner. By working with industry experts in the field of manufacturing, it was proposed to utilize injection molding simulation software to develop the process windows so that production time is not wasted from experimentation. This work focuses on constructing a new methodology in which manufacturers can modify the process window to target issues they commonly experience during production. By utilizing this approach, manufacturers can prevent and identify defects before a part has begun production.

## 2. Materials and Methods

### 2.1. Materials

The injection molding CAE software package chosen to use in this project was Autodesk Moldflow Synergy 2023, product version 45.2.60.60. Moldflow was chosen to provide an easy-to-use methodology for the research sponsor of this project since they are already familiar with the software. Moldflow includes a database of most of the common thermoplastic materials from producers and distributors. The database of the software provided a material profile for the high-impact polystyrene (HIPS) thermoplastic material of Styron 478 from Americas Styrenics. This material was characterized by rheology, PVT, mechanical, and thermal testing. Styron 478 was selected because it is an amorphous material, and it was desired to eliminate the effect of crystallinity in the analysis, in particular shrinkage and warpage [18]. Furthermore, the properties of semi-crystalline material are more difficult to predict using simulation software.

A 180-ton Sumitomo injection molding machine (SG180M-HP, Tokyo, Japan) was used along with the physical ASTM mold to produce parts used in the comparison against results from simulations. The ASTM mold was chosen since it is complex and readily available for experimentation. It consists of 4 non-uniform cavities, each producing a specimen used for the testing of moldable materials according to the following ASTM standards of testing plastics: D638-22 for tensile strength testing, D540-21 for impact testing, and D790-17 for flexural testing [19,20,21]. Four pressure sensors, type 6157C from Kistler, were installed into the cavity of the B half of the ASTM mold. The four sensors allow for the collection of injection pressure during filling and pressure during packing at each location for the duration of the production cycle. A ComoNeo from Kistler located in Novi Michigan in the United Stated is connected to the pressure sensors to collect data every 0.028 s.

### 2.2. Model Development

To develop a model for simulation, a company was hired to conduct a 3D scan of the cavity of the ASTM mold. The scan was reverse-engineered into a model suitable for simulation. Additional features of the mold such as the housings, water lines, and sprue were measured and generated using SolidWorks 2024. The completed model was discussed with representatives from Moldflow. It was with their recommendation that to define the system boundaries, the model be cut to only include the A half of the mold, a portion of the B half, which included the cavity, cooling lines, and a portion of the mold housing. Both the original and the cut models of the ASTM mold are shown in Figure 1. After cutting the model, heat convection with the software’s recommended heat transfer coefficients was applied as the boundary conditions. By doing so, the system of the model was defined, and the boundary was established [22,23].

The reduced model was imported into Moldflow with each portion of the mold being attributed as either a part, water line, mold base, or mold insert based on their experimental function. By assigning portions of the mold attributes, Moldflow is able to distinguish their function as the model is analyzed. Each portion of the model was then meshed using Moldflow’s 3D solid style of meshing at the default size recommended by the software for each portion’s corresponding attribute. Meshing each portion of the model based on the attribute, such as the part itself, requires different mesh sizes so that the model can be appropriately simulated. In total, the model consisted of 3,890,164 elements. Additionally, four sensor nodes were placed in the model at locations corresponding to the pressure sensor locations in the ASTM mold. The nodes allowed pressure data at these locations to be collected and compared against experimental results [22]. Figure 2 displays the locations of the four pressure sensors.

The model of the ASTM mold was modified with the assistance of representatives from Moldflow. During experimentation, it was noticed that a tip of material at the end of the sprue was being produced by the nozzle of the injection molding machine. Since the original scan of the ASTM mold did not include the nozzle, the tip was not included in the ASTM model. An updated model was developed to include this tip of material and used subsequent simulations for which data are presented [22]. Figure 3 depicts the updated ASTM model.

### 2.3. Development of Process Windows

The main objective of this work is to demonstrate the use of CAE to develop injection molding process windows. This objective will be accomplished by utilizing the ASTM mold as an example to develop a method to construct the process windows. Before working towards this objective, the software’s initial predictions were compared against analytical solutions to ensure the results made sense [22]. Choosing a combination of CPVs in the area depicted by the graph in a process window will result in the manufacturing of an acceptable part. Combinations of CPVs outside of this window result in a defect in the part. Process windows developed through simulation can give an operator the opportunity to identify these conditions without wasting time on experimental trial and error. The development of process windows is split up into two stages: filling and packing.

An important aspect in the manufacturing of a part is to understand how a part will fill and the challenges that can occur in this stage. Figure 4 depicts how the ASTM part fills both experimentally and in the predictions of the simulations. The process window for the filling stage is developed as a function of the three CPVs injection time, melt temperature, and mold temperature. The goal is to identify combinations of these CPVs that produce an acceptable part. The filling-stage process window utilizes a response of the maximum injection pressure experienced during production from a resulting fill time chosen for a combination of CPVs. In total, 198 combinations of the three CPVs were tested to collect data to construct this process window from the selections of 3 melt temperatures, 3 mold temperatures, and 22 fill times. In total, 22 fill times were chosen to test the entire range of a part’s production for both low fill times, which will be beneficial to manufacturers, and high fill times to determine if the software is able to predict points at which a non-fill occurs.

Packing, the second stage this work focuses on, occurs next as the screw applies a constant pressure, known as the packing pressure, to the melt within the cavity of the mold. This causes the material to expand towards the walls of the mold manufacturing a full part while minimizing part warpage and shrinkage. The process window for the packing stage is developed as a function of the three CPVs of packing pressure, mold temperature, and melt temperature. The packing stage process window is developed using shrinkage % as a response to the resulting packing pressure applied. For the case of the ASTM model, shrinkage % results were collected from the thickness of the thick flex test bar since it is the portion of the part with the largest thickness, meaning it will be most impacted by shrinkage [22]. The selection of 3 melt temperatures, 3 mold temperatures, and 9 packing pressures, ranging through the capabilities of packing pressure the ASTM mold, can withstand for experimentation with the selected injection molding machine, resulting in 81 combinations of CPVs being tested to collect the data necessary for the construction of this window.

### 2.4. Selection of Simulation Processing Parameters

Combinations of the three selected CPVs were investigated to build the process window for the filling stage. The three controllable process variables were injection time, melt temperature, and mold temperature. Injection times of 0.5, 0.75, and 1–20 s, evaluated every second, were chosen as simulation parameters. Although the tested injection times are longer than would be acceptable in manufacturing facilities, they were evaluated for a comparison between the software and experiments at points at which a non-fill is produced. Three melt temperatures of 200 °C, 220 °C, and 240 °C were selected based on the range of processing temperatures of the HIPS Styron 478 material [20]. The material manufacturer recommends the lower processing temperature limit to be 200 °C and the upper processing temperature limit to be 240 °C with 220 °C being in the middle of this range. Finally, the mold temperatures chosen for the simulations were selected using the same reasoning with temperatures of 30 °C, 45 °C, and 60 °C [20].

Following the same approach, combinations of CPVs were investigated to construct the process window for the packing stage. The three CPVs were packing pressure, mold temperature, and melt temperature. Values for the CPV of packing pressure were chosen by focusing on the evaluation of the process window. The injection molding machine used for experimentation applies packing pressure (in psi) according to the machine’s hydraulic pressure. Experimentation of developing process windows using the injection molding machine would have resulted in testing nine equidistant values from the range of 50 psi–450 psi [22].

As a means of correlating the packing pressures used for experimentation to simulation, a conversion of pressure in psi from the machine’s hydraulic pressure to MPa was used. The conversion came from previous experimentation. Values and profiles of the pressure collected from the sensors located in the mold were plotted with the values and profiles obtained through the simulation’s sensor node predictions. Once plotted on the same graph, the means of correlating the two values was established [22]. For example, a machine hydraulic pressure of 15,000 psi converts to 67.5 MPa in the software. Therefore, nine packing pressures ranging from 6 MPa to 62 MPa were evaluated as parameters for the simulation. Each of the nine packing pressures was applied constantly to the part for the duration of the packing stage. The development of the packing stage process window utilized the same values for melt and mold temperature as the filling-stage process window.

The remainder of the process parameters required for the simulation and the analysis type were chosen and kept constant between each of the simulations. To resemble the experimental process, each simulation in Moldflow was performed using an analysis sequence of Cool (FEM) + Fill + Pack + Warp. This analysis sequence allows the software to simulate a cooling stage before molding, the filling stage, the packing stage, as well as the warpage of a part after ejection once it cools to room temperature. The Cool (FEM) portion of the analysis occurs before the filling stage begins where the temperature in the mold is calculated using FEM (Finite Element Method). This portion of the analysis allows the mold to reach a quasi-steady state before material is injected into the cavity.

A conduction time of 50 s for the ASTM mold was estimated using the following formula [24]:(1)$$t_{CT}=\frac{h^{2}}{\alpha }$$
where$t_{CT}=Cooling\;Time$h=half the part’s thickness$\alpha =Thermal\;Diffusivity=\frac{k}{\rho C_{p}}$k=Thermal Conductivityρ=Density

The conduction time consists of two components. The packing time is set to 20 s and the cooling time when packing pressure is not applied is set to the be 30 s. This separation within the conduction times comes from previous experimental testing of the ASTM mold where it was found that a ratio of 40% packing time to 60% cooling time where packing pressure is not applied provided improved mechanical properties for the final part. Packing at a ratio larger than 40% of total conduction time resulted in the final part having less tensile stress at both yield and break along with less tensile strain at break [17].

During the 20 s of packing time for the filling-stage process window, a packing pressure of 27.6 MPa was used. This value comes from the middle of the range for packing pressures tested for the development of the packing stage process window. For the packing stage process window, a fill time of 2 s was chosen as a constant parameter to ensure that an established fill time will not impact the results of the simulations. Furthermore, a mold open time of 7.1 s and a mold close time of 12.1 s were selected based on data collected from experimental runs.

### 2.5. Experimental Validation Methodology

To validate each of the process windows constructed through simulation, the following experimental methodology was followed. The barrel and mold began heating up to the desired temperature 45 min before any trials were conducted. After both had reached the desired temperature, the barrel was purged five times to allow the material to reach a consistent state for molding as well as to remove any material from previous runs. Five cycles were run to produce parts that were discarded so the mold could reach a quasi-steady state. At this point, data collection began for each combination of the processing variables tested. Ten samples were collected for each combination. In the cases where multiple combinations of processing variables were tested in the same run, data for the first five trials were not collected between the changing of variable settings.

## 3. Results and Discussion

### 3.1. Filling Stage Simulation Results

To construct the filling-stage process window, the first step is to simulate each possible combination of CPVs for the ASTM mold in the simulation software. In each simulation, the maximum injection pressure experienced by the model is collected. The values of maximum injection pressure from the simulations according to the combination of injection time, melt temperature, and mold temperature were chosen. The maximum injection pressure collected for each simulation comes from the point of injection specified in the simulation model. In the case of the ASTM mold, it is located at the very end of the sprue. Previous testing has been conducted using the pressure sensors located both in the cavity of the mold and the ASTM simulation model. In this research, predicted and experimental pressures experienced by both were compared and analyzed [22]. Figure 5 displays the maximum injection pressures for a melt temperature of 200 °C.

In general, as the fill time increases, the fill pressure decreases until the time when the increase in viscosity due to the cooling from the mold walls becomes significant and the fill pressure begins to increase. This effect is noticeable mostly at lower mold temperatures, which results in higher injection pressures. Furthermore, it can be seen in Figure 5 that the data stop at 9 s for a mold temperature of 30 °C, 12 s for a mold temperature of 45 °C, and 17 s for a mold temperature of 60 °C. After each of these points, the software predicts a non-fill due to the cooling from the mold walls. Similar trends are experienced with simulations using combinations of CPVs at melt temperatures of 220 °C and 240 °C. Data are shown in the Appendix A.

### 3.2. Filling-Stage Process Window Development

Development of the process window for the filling stage involved using the data collected from the simulations to construct a window-shaped area on a graph. This window-shaped area visually indicates which combinations of controllable process variables will produce an acceptable part. To construct this window, boundaries of the processing limits need to be established. The boundaries of the processing limits can be selected depending upon the desired performance measures of the manufacturer including the elimination of particular defects or adherence of parts critical to quality dimensions. In the case of the process window developed for the ASTM mold, the boundaries consist of high and low fill times resulting in the production of an acceptable part. These limits will establish a point at which combinations of controllable processes result in a non-fill due to cooling or the maximum injection pressure predicted during the simulation exceeds a set limit.

The lower boundary processing limit is influenced by the maximum injection pressure predicted by a simulation for a particular injection time. Too large of an injection pressure in a process can result in the part flashing or causing the injection molding machine to pressure limit resulting in a non-fill. A part produced by the ASTM mold which experienced flash is shown in Figure 6. Depending on an injection molding machine’s capability, the lower boundary processing limit can be established at a known point from prior production. By selecting a point below this threshold, a processor can ensure that the process will adhere to previous standards.

The upper boundary processing limit is influenced by the allowable fill time of a process. When a process takes too long, then a short shot will be produced due to the cooling from the mold walls. The upper boundary processing limit can be set to avoid a short shot being produced. An example of a short shot is shown in Figure 7. In some cases, it is beneficial to part quality to have process windows with longer cycle times even at the expense of cycle time. Faster injection times introduce higher stress in part and can cause defects or burns on the part from poor venting [17]. The specific choice of the maximum time is up to the discretion of the processor and will vary from part to part depending upon its requirements for production.

Using the established boundary limits of a complete part with no maximum injection pressure, a process window for the filling stage developed from the simulation’s data is shown in Figure 8. Figure 8 provides a graphical representation of the combinations of fill time, melt temperature, and mold temperature which will result in the manufacturing of an acceptable part. Areas outside of the marked regions will result in the manufacturing of an unacceptable part, which will experience one of the aforementioned defects. Additionally, a processor can utilize the process window in Figure 8 to determine the maximum injection pressure expected from a chosen combination of CPVs by looking at the data presented in Figure 3.

The boundary processing limits can be adjusted to the required specifications of a previously acceptable process. For the version of the process window shown in Figure 4, there is no limit for maximum injection pressure and the amount of fill time is limited by a short shot being produced due to cooling from the mold walls. A different process window can be developed by modifying these parameters while utilizing the same data. Figure 9 depicts a process window for the filling stage in which the lower boundary processing limit has an allowable maximum injection pressure of 45 MPa. A comparison of the two process windows shows that the area in which an acceptable part would be produced is reduced with the new limitations. With this limit, a melt temperature of 200 °C will no longer produce an acceptable part with portions of the areas of other melt temperatures being reduced as well.

In the case of constructing a process window for the filling stage for the ASTM mold for experimental evaluation, different boundaries were used. As previously mentioned, the injection molding machine used for experimentation was a 180-ton Sumitomo machine. The fill peak hydraulic pressure limit established on this machine before experimentation began was 15,000 psi [22]. When this pressure limit is reached, the machine immediately switches from the filling stage to the packing stage producing a short shot. Based on the pressure sensor output, the estimated injection pressure equivalent to a hydraulic pressure of 15,000 psi is 67.5 MPa [22]. For the upper boundary processing limit, the allowable fill time is determined by a short shot being produced due to the cooling of the mold walls. Although the acceptable fill times are very large, shorter fill times can be used within the window. The process window constructed using these processing limits is shown in Figure 10.

### 3.3. Filling Stage Experimental Validation

The next step of the project is to experimentally validate the recommended combinations of CPVs depicted by the area of the process window presented in Figure 6. Following the same experimental procedure listed previously, trials consisting of several combinations of controllable process variables were performed. Data collected in each trial includes responses from the injection molding machine consisting of the actual fill time (s), fill peak pressure (psi), V/P switch position (in), and cushion position (in). Table 1 displays the data collected for trials consisting of a melt temperature of 200 °C and a mold temperature of 30 °C. At the set fill time of 0.75 s, a short shot was produced due to the machine reaching the established fill peak pressure limit. The machine also begins to pressure limit at a set fill time of 1 s, causing instability in the part’s production. This finding aligns with the process window for the filling stage as it predicts an unacceptable part being produced at this combination of CPVs.

Figure 11 is developed by plotting the fill peak pressure data from experiments shown in Table 1 along with the maximum injection pressure values predicted by the simulations shown in Figure 3. This graph was constructed by selecting a data point from both the simulations and the experiments collected at the same fill time. At this chosen point, the data presented for the fill peak pressure was scaled until it matched the same location as the data collected from the simulation. By matching this point, the two outputs from the data sets can be compared. The graph in Figure 11 shows the smallest and largest fill times resulted in the highest pressure both experimentally and in the simulation’s predictions. In Figure 11, it appears that the maximum injection pressures predicted by Moldflow seem to agree with the fill peak pressures collected through experimentation which had a standard deviation of 0.0055 s, thus providing confidence in the experimental validation of the process windows.

Experimental validation of the process windows developed through simulations results in the following conclusions. The first conclusion is that the process window made through simulation is acceptable with all regions within the window producing an acceptable part. Regions outside of the process window will either produce a part with defects or exceed the machine injection pressure limit. Additionally, trials from experimentation show that the maximum injection pressures during fill are similar to the fill peak pressure measured by the injection molding machine. Therefore, the methodology of developing a process window through the use of CAE can be utilized within a manufacturing facility as an acceptable approach.

### 3.4. Packing Stage Simulation Results

Shrinkage %, the targeted response for developing the packing stage process window, is obtained within the simulation for each unique CPV combination by examining the warpage results. Within this results tab, a color map containing the part’s total warpage after the part has been ejected from the mold cooled to room temperature can be seen. Two nodes are selected in the desired dimension, in this case, the thickness of the thick flex test bar, to obtain the shrinkage % of the selected study. The two nodes calculate the distance that a part has shrank from its position in the original ASTM model [22]. The nominal shrinkage for the material used in this study is 0.5844% [25].

The thickness shrinkage % from the thick flex test bar collected using this method is shown in Figure 12. Figure 12 displays the predicted shrinkage % of this dimension from simulations with varying packing pressures, mold temperatures, and a melt temperature of 200 °C. The data within this figure show that at each packing pressure value, the highest mold temperature of 60 °C results in the largest shrinkage %. Additionally, as packing pressure increases, shrinkage % decreases across the three mold temperatures until a point at which the values become constant. At this point, increasing the packing pressure no longer has an impact on shrinkage. The shrinkage % data collected at melt temperatures of 220 °C and 240 °C depict similar trends to the data collected at a melt temperature of 200 °C (Appendix A).

### 3.5. Packing Stage Process Window Development

The shrinkage % of the thickness of the thick flex test bar collected from the simulation’s predictions was used to construct the packing stage process window. The lower and upper boundary processing limits were established based on the shrinkage % result. Similar to the development of the filling-stage process window, the boundaries of the packing stage process window can be manipulated to adhere to a manufacturer’s desired performance measures of a part. The lower boundary processing limit is influenced by defects in a part produced with too low of a packing pressure. Too low of a packing pressure can result in a response of a high shrinkage impacting the final dimensions of a part thus impacting a critical quality dimension, making it unacceptable. Using a process where the packing pressure is too low can also result in sinks, warpage, and short shots, all of which are unacceptable defects. An example of a sink in the ASTM part is shown below in Figure 13. To avoid these outcomes, the lower boundary limit is established by choosing packing pressure with a correlating shrinkage % for which a defect is not predicted by the simulation.

The upper boundary processing limit is influenced by having too high of a packing pressure within a process which can result in the part not experiencing enough shrinkage. Two outcomes can occur with a part experiencing too little shrinkage. Just as in the lower boundary processing limit, a part may become unacceptable if a part’s dimension is too large to meet a specified requirement. This requirement comes from a dimensional tolerance of the part which is critical to quality and the customer. The second outcome is the part can become lodged within the B half of the mold during ejection. Failure to eject a part can result in damage to the part and the tool as well as the loss of cycle time. In a possible scenario where a part is not ejected in a complex tool consisting of lifters and slides, the part can shift and could cause damage to these components and the part itself as the mold attempts to close. Taking these two outcomes into consideration, the upper boundary processing limit is established based on the dimensional tolerances of a part as well as packing limits for a machine’s capability.

A process window for the packing stage can be constructed by selecting the desired amount of maximum and minimum shrinkage % to generate the upper and lower boundary processing limits. Figure 14 displays an example of a process window that can constructed in this manner by using the limits of a maximum shrinkage of 0.95% and a minimum shrinkage of 0.85%. Figure 14 provides a graphical representation of the possible combinations of packing pressure, mold temperature, and melt temperature which will produce a part with a shrinkage % of the thick flex test bar between these two limits. In this process window, a combination of CPVs that are outside of the marked region will result in a part whose shrinkage % does not meet these criteria, making it unacceptable. Within the process window, the area on the lower left-hand side of the graph is marked with a dotted line. The dotted line notes that if a combination of CPVs consisting of a melt temperature of 200 °C, and a packing pressure less than 13.8 MPa is chosen, then a short will be produced instead of the part not meeting the established requirements.

To develop a process window for an experimental evaluation of the ASTM mold, different boundaries were used. When viewing the average volumetric simulation results within the simulation, a sink appears in the center of the thick flex test bar at lower packing pressure values [22]. Since the sink is a visible, major defect, its appearance will provide the limits for the lower boundary of the process window. The sink appears when the measured shrinkage in the simulation is above 1.02%, therefore the lower boundary processing window is established at this point [22].

The data presented in Figure 12 reveals a point at which the shrinkage % becomes constant at each tested melt temperature. At these points, the parts no longer shrink even as additional packing pressure is applied. Since applying additional packing pressure at these points is no longer beneficial, the upper boundary processing limit is established at the amount of packing pressure before the resulting shrinkage becomes linear for each combination of CPVs. Furthermore, setting the upper limit at these points provides the added assurance that the parts that do not experience enough shrinkage do not get stuck or damaged during ejection. The practical application of the ASTM mold process window for the packing stage using these two boundaries is shown in Figure 15. This process window follows the same notation as the previous example process window in Figure 14.

### 3.6. Packing Stage Experimental Validation

Using the established methodology of producing experimental parts for validation, the thick flex test bars produced were measured with a dial caliper. The thickness of the bar was measured in the center of the part at the corresponding location data from the simulations was collected. An average of the measurements was taken across 10 replicates for each trial, with an exception for experiments where parts became damaged during ejection. The measurements were converted into shrinkage % using the equation below where the actual dimension is based upon the thickness of the scanned CAD model of the part:(2)$$Shrinkage=\frac{\left(Actual\;Dimension-Measured\;Dimension\right)}{Actual\;Dimension}$$

Table 2 presents the shrinkage % data collected for the packing stage evaluation. Table 2 consists of two combinations of melt and mold temperatures used within the process window (200 °C melt temperature/30 °C mold temperature and 240 °C melt temperature/60 °C mold temperature) to evaluate the extreme points of the process window parameters. Different values of packing pressure, in both psi according to the hydraulic pressure of the machine and its corresponding conversion to MPa in Moldflow, were tested for each mold and melt temperature combination. From each of these combinations, the experimental shrinkage % was measured and it was noted whether or not the part failed to eject from the mold.

Within Table 2, the shrinkage % measurements from experiments were higher than the shrinkage % predicted by the simulation software. Although these values do not equate, Moldflow was able to predict the shrinkage % values within 1% throughout each packing pressure tested with a root mean square error of 0.00318. The discrepancies in the values in the predictions of the simulations and the experimental trials could occurred because of limitations of the software, unknown heat transfer coefficients, or the differences between experimental conditions and the standard boundary conditions used within the model. Additionally, the simulation software predicts a similar trend to what was found in experiments. In both sets of data, the shrinkage % decreases as higher amounts of packing pressure are applied to the part until a point at which the shrinkage % becomes constant. After this point, applying additional packing pressure no longer has an effect on the shrinkage %. This implies that the upper limit for the process window construct for the ASTM mold can be established using this indication.

Furthermore, from the combinations of CPVs tested, only the combination of 240 °C melt temperature, 60 °C mold temperature, and a packing pressure of 55.2 MPa resulted in the part getting stuck during ejection. This combination of CPVs is outside of the marked region of the process window shown above in Figure 10. Additional experiments were also conducted utilizing a 200 °C melt temperature, 30 °C mold temperature, and a packing pressure of 6.9 MPa. When producing parts utilizing this combination of CPVs, a sink appeared in the center of the thick flex test bar, just as the simulation predicted [22]. Once again, this combination of CPVs is outside of the marked region of the process window, thus providing credibility that the process window is properly constructed through simulation.

Experiments of the boundaries of the process window developed in simulation lead to the following conclusions. First, the process window made through simulation consists of all regions within the window-producing part without a sink or being stuck inside the mold. A combination of CPVs outside of the process window can result in a defect. Additionally, there is a less than 1% difference between the experimental shrinkage % data and the simulations’ predictions along with similar trends. This results in the discovery that the simulation is able to successfully construct a process window for the packing stage using simulation shrinkage % results. In future cases, the packing process window can also alter a desired part’s production by eliminating specified defects or achieving critical part quality tolerances.

## 4. Conclusions

This project focused on developing a method to construct process windows using injection molding simulation software with the process windows split up into two stages: filling and packing. Three controllable process variables were chosen to develop the process window in each stage and are shown in Table 3 below. The filling-stage process window was developed by using a response of maximum injection pressure and fill. Utilization of these two responses allowed for the process window to be easily manipulated to adhere to production requirements. In this study, three filling-stage process windows were developed, all using the same data, to demonstrate examples of how they can be modified to a requirement or performance measure. Furthermore, process windows for the packing stage were developed using a response of shrinkage %. Once again, processors can utilize data collected from the CAE’s predictions to construct numerous process windows, each with their own specifications and adherence to desired performance measures. These process windows can also be constructed before a mold has entered their facility, giving processors an advantage in knowing how to correctly manufacture a part without wasting production time or materials.

Process windows developed through CAE data for both the filling and packing stage underwent experimental validation. In both process windows, experimental trials revealed the processing limits for each desired performance measure aligned with the limits established by the process window. The areas captured by the process windows were shown to include all combinations of CPVs which resulted in the manufacturing of an acceptable part according to the standards set in place. Additionally, the methodology developed in this work presents the opportunity for manufacturers and future researchers hoping to develop process windows for their own parts. Therefore, this work establishes trust in the simulation software’s ability and its utilization as a tool to simulate production and develop process windows for manufacturers.

## 5. Future Work

After reviewing this work with our industry partners, several avenues for future projects were recommended to focus on how to optimize this process to be beneficial for manufacturers. The first area recommended for future work is to investigate simpler methods of simulating the ASTM model. In this work, each simulation testing a combination of CPVs took an average of 1 h and 30 min to complete. Manufacturers do not have this amount of time to diagnose and solve an issue during production. Therefore, it is the goal of this new project to reduce the amount of time the simulations take to complete while reaching the same outcome of successfully developing a process window.

Other recommendations for future work include applying the methodology constructed in this paper to a part chosen by our sponsor. Since the development of the process windows is focused on the defects pertaining to the ASTM part, CPVs other than injection time, packing pressure, melt temperature, and mold temperature will be chosen. Furthermore, analyzing a new part provides an opportunity to use this method with other types of polymers. The selection of new CPVs will serve as an evaluation of the method developed in this work to analyze if similar conclusions can be reached. It is recommended in this research to perform a DOE to optimize the selection of CPVs which will most impact the appearance of defects. The DOE will also allow for the reduction in the number of simulations required to develop a successful process window.

## Figures and Tables

**Figure 1 polymers-17-00718-f001:**
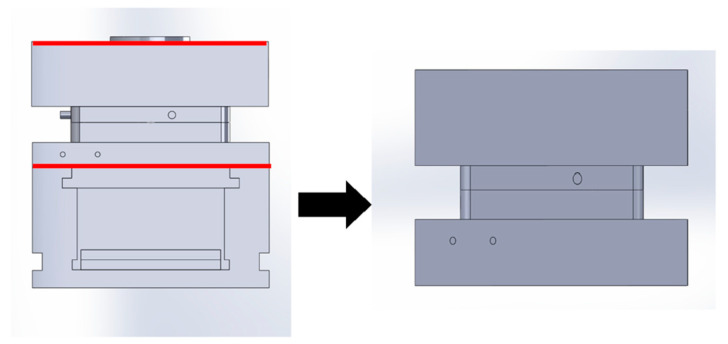
Original and cut ASTM model.

**Figure 2 polymers-17-00718-f002:**
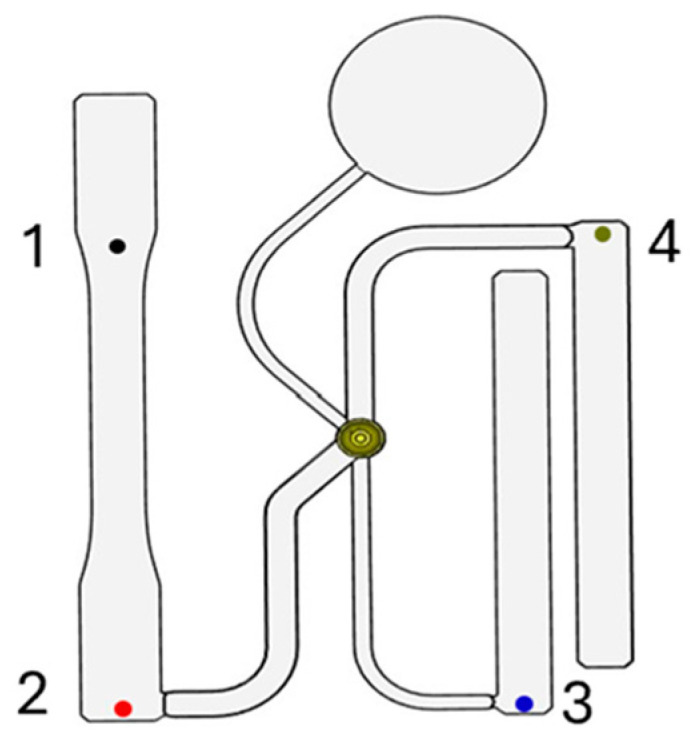
Four pressure sensor locations.

**Figure 3 polymers-17-00718-f003:**
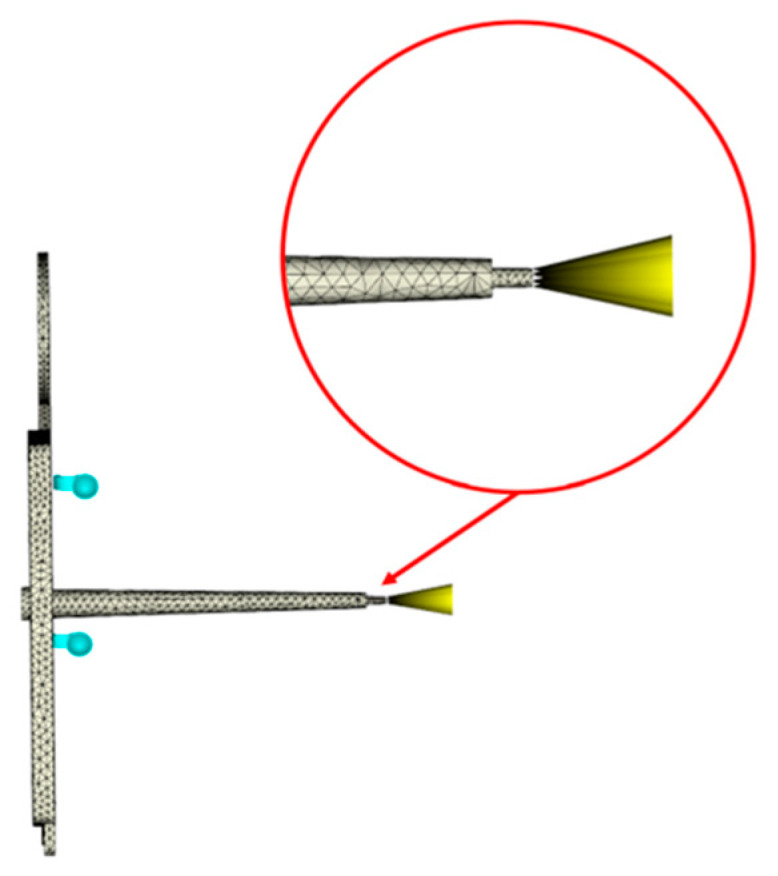
Updated ASTM model.

**Figure 4 polymers-17-00718-f004:**
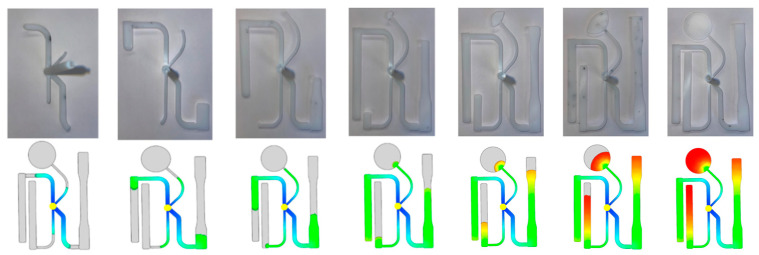
Fill pattern comparison of experimental trials and Moldflow.

**Figure 5 polymers-17-00718-f005:**
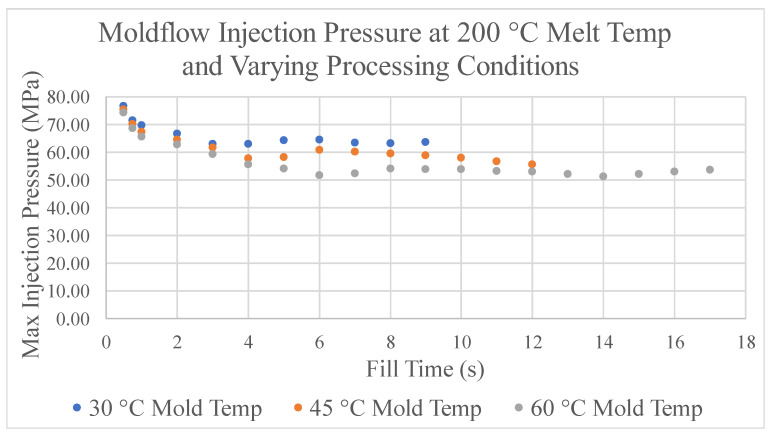
Moldflow injection pressures at 200 °C melt temp and varying process conditions.

**Figure 6 polymers-17-00718-f006:**
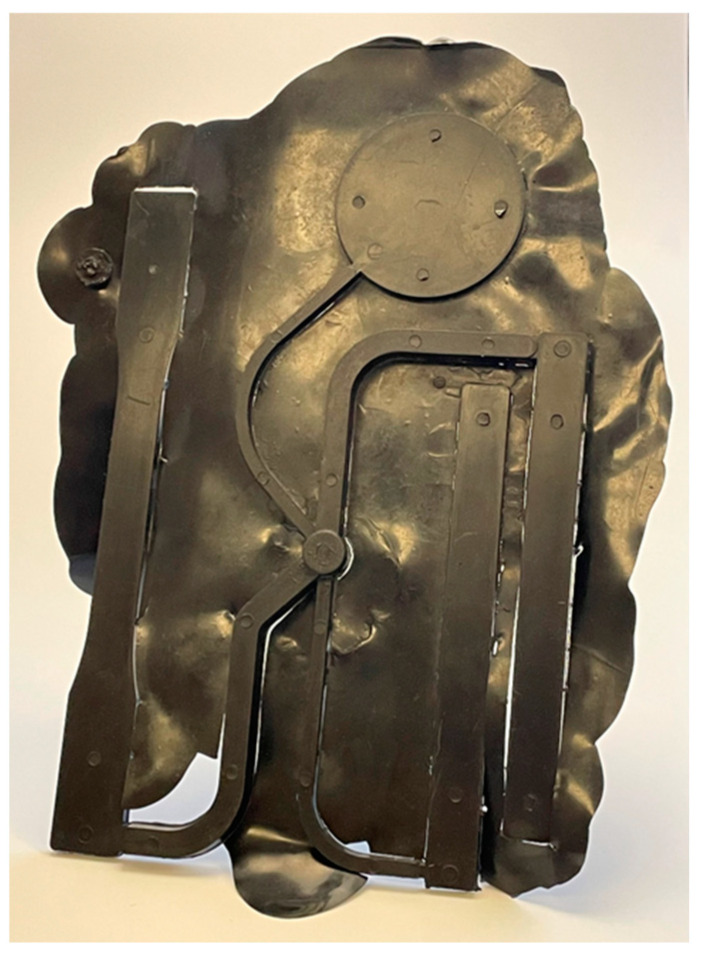
ASTM part with a flash defect.

**Figure 7 polymers-17-00718-f007:**
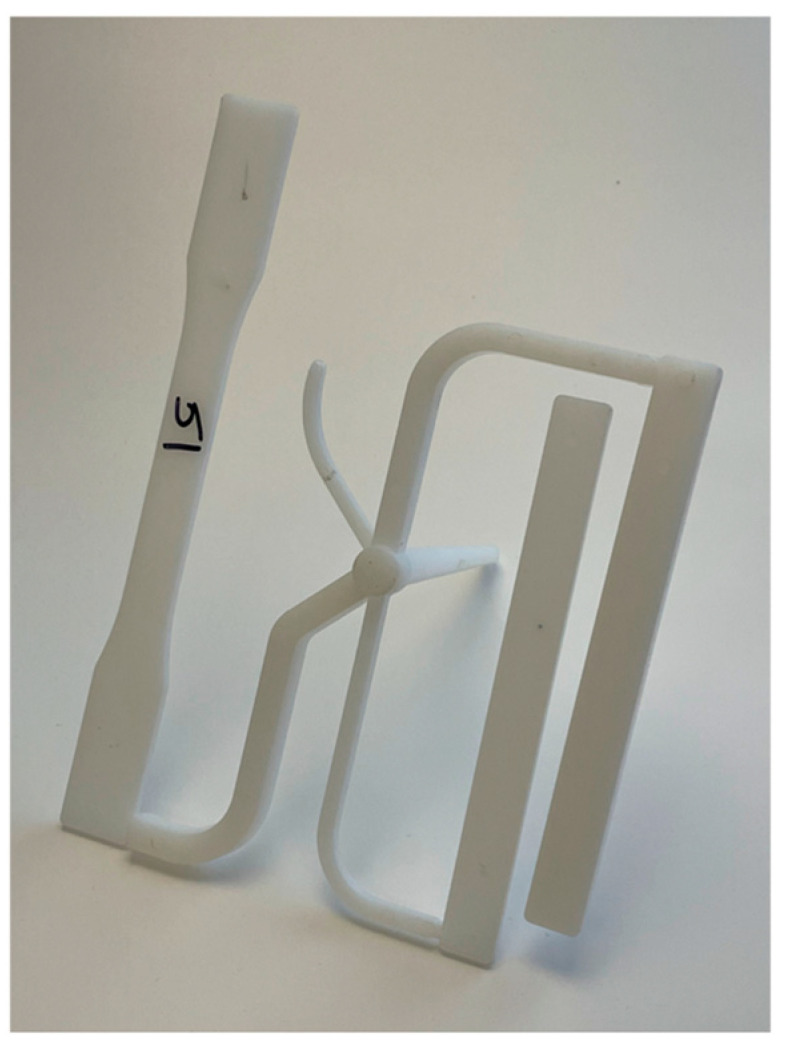
ASTM part with a short shot.

**Figure 8 polymers-17-00718-f008:**
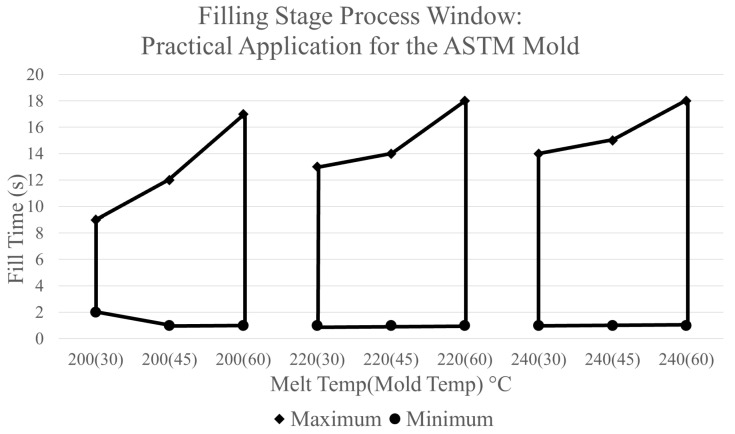
Filling-stage process window—producing full parts/no injection pressure limit.

**Figure 9 polymers-17-00718-f009:**
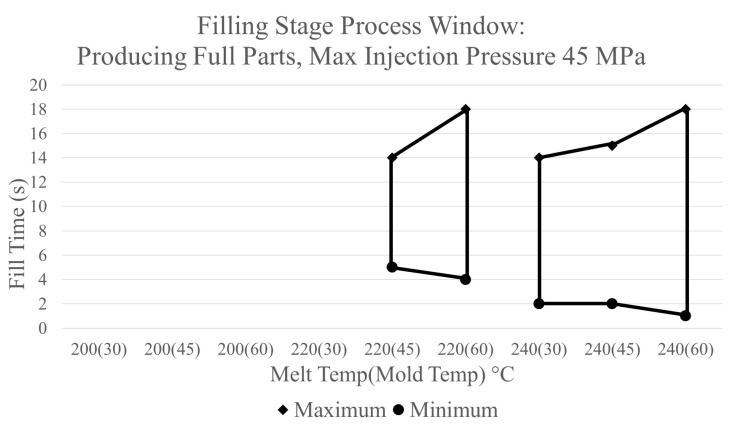
Filling-stage process window—producing full parts/injection pressure limit of 45 MPa.

**Figure 10 polymers-17-00718-f010:**
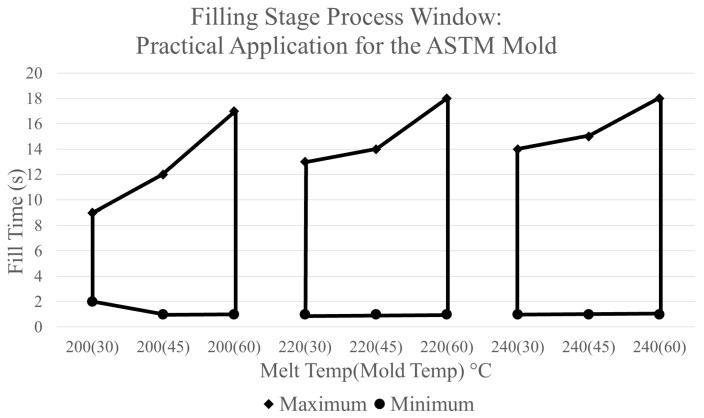
Filling-stage process window—practical application for the ASTM mold.

**Figure 11 polymers-17-00718-f011:**
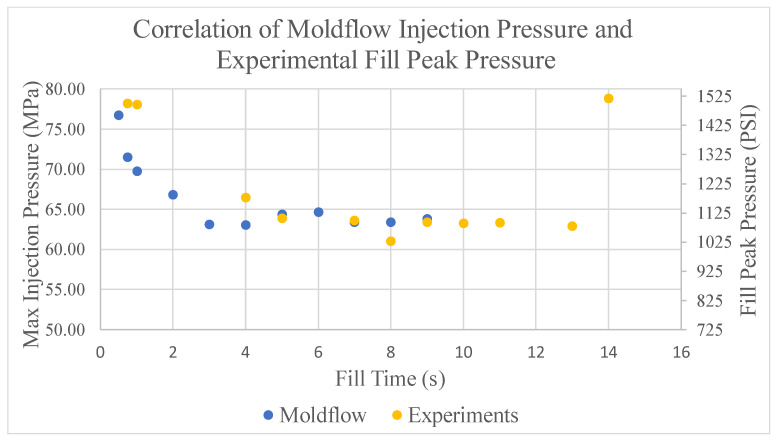
Correlation of Moldflow injection pressure and experimental fill peak pressure data.

**Figure 12 polymers-17-00718-f012:**
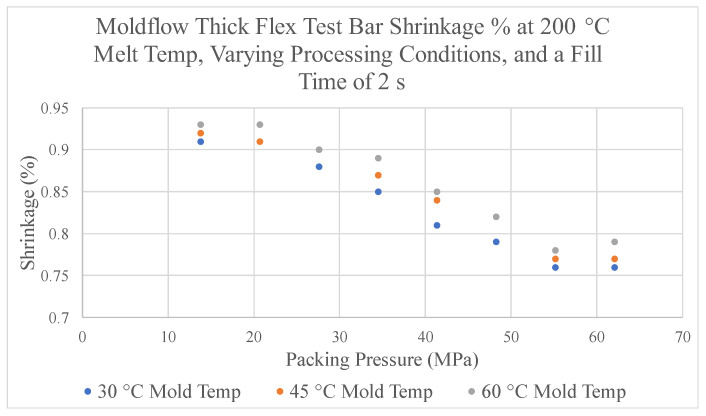
Moldflow thick flex test bar shrinkage % at 200 °C melt temp and varying process conditions.

**Figure 13 polymers-17-00718-f013:**
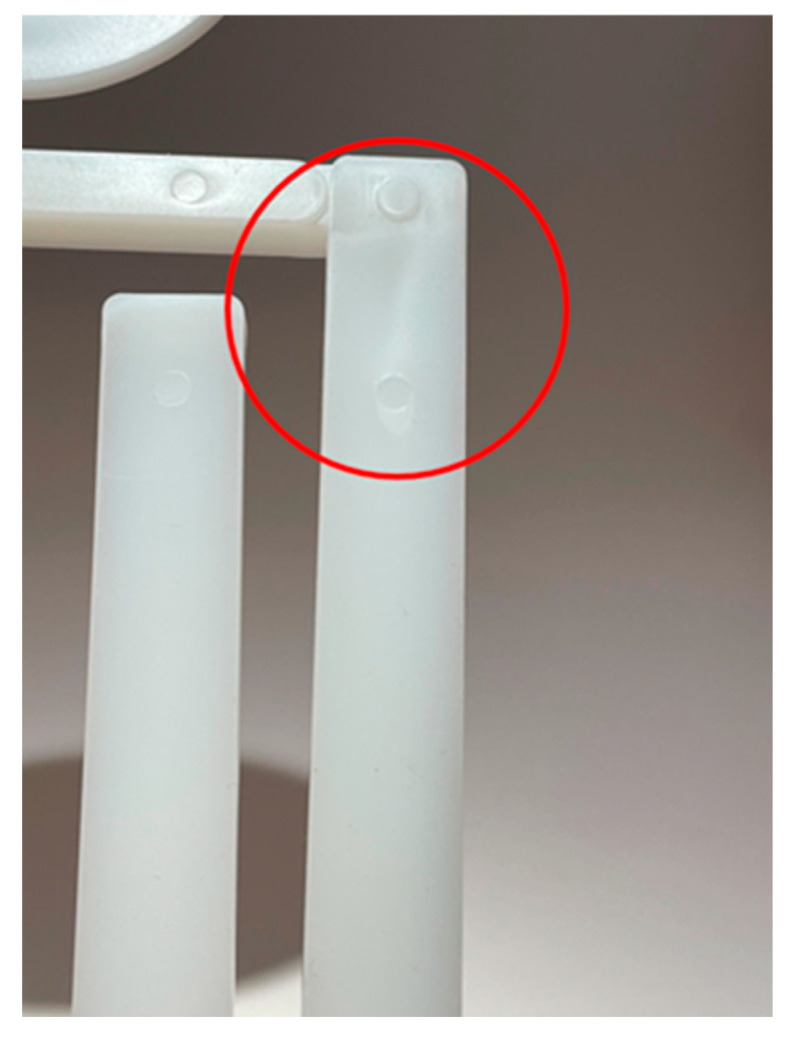
ASTM part with a sink defect.

**Figure 14 polymers-17-00718-f014:**
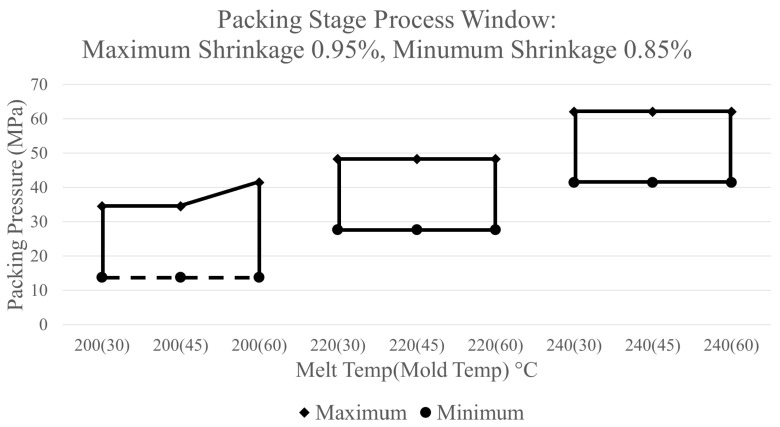
Packing stage process window—max shrinkage of 0.95%, min shrinkage of 0.85%.

**Figure 15 polymers-17-00718-f015:**
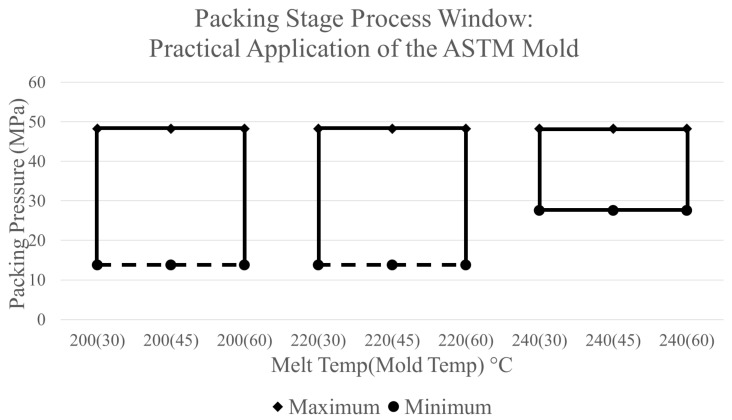
Packing stage process window—practical application of the ASTM mold.

**Table 1 polymers-17-00718-t001:** Average experimental data for filling stage at 200 °C melt temperature and 30 °C mold temperature.

Set Fill Time (s)	Avg Actual Fill Time (s)	Avg Fill Peak Pressure (psi)	Avg V/P Switch Position (in)	Avg Cushion Position (in)	Did the Part Short?
0.75	0.195	1481	1.557	1.502	Yes
1	0.954	1477	0.217	0.207	No
4	3.733	1157	0.209	0.207	No
5	5.058	1088	0.209	0.208	No
7	7.060	1079	0.209	0.209	No
8	7.976	1008	0.209	0.209	No
9	9.137	1073	0.209	0.209	No
10	10.271	1069	0.209	0.209	No
11	11.184	1072	0.209	0.209	No
13	12.961	1061	0.209	0.209	No
14	14.035	1498	0.276	0.276	Yes

**Table 2 polymers-17-00718-t002:** Shrinkage % data for packing stage evaluation.

Melt Temp (°C)	Mold Temp (°C)	Experimental Packing Pressure (psi)	Moldflow Equivalent Packing Pressure (MPa)	Experimental Shrinkage %	Experimental Shrinkage % Standard Deviation	Moldflow Shrinkage %	Part Stuck During Ejection?
200	30	100	13.8	1.39	0.052	0.91	No
200	30	150	20.7	0.99	0.053	0.91	No
200	30	350	41.4	0.87	0.053	0.79	No
200	30	400	55.2	0.87	0.042	0.76	No
200	30	450	62.1	0.87	0.063	0.76	No
240	60	50	6.9	1.59	0.063	1.02	No
240	60	100	13.8	1.23	0.000	1.06	No
240	60	150	20.7	1.37	0.057	1.04	No
240	60	200	27.6	1.19	0.053	1.01	No
240	60	400	55.2	0.40	0.067	0.90	Yes

**Table 3 polymers-17-00718-t003:** Controllable process variables chosen for each process window development stage.

Process Window	Controllable Process Variable
Filling Stage	Injection Time (s)
	Melt Temperature (°C)
	Mold Temperature (°C)
Packing	Packing Pressure (psi)
	Melt Temperature (°C)
	Mold Temperature (°C)

## Data Availability

Data available from the authors.

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
