# Peer review of "Utilizing Simulation Software to Develop Injection Molding Process Windows with High-Impact Polystyrene"

_polymers, 2025, doi:10.3390/polym17060718_

Round 1
Reviewer 1 Report
Comments and Suggestions for Authors
In this work, authors developed methodology for the construction of process windows by testing controllable process variables in the filling and packing stage of the injection molding process. The process window was experimentally assessed to establish confidence in the software. Before considering publication, there are several critical points and potential areas for improvement in the manuscript:
- The study focuses on utilizing simulation software for developing injection molding process windows, which is a well-explored topic in previous literature. While the methodology is useful, the paper does not clearly highlight what is novel about the approach or how it differs from prior work. In this regard, authors should emphasize unique contributions, such as specific improvements in methodology, new experimental validation techniques, or enhanced prediction accuracy.
- The study does not extensively discuss the limitations of using simulation software to develop process windows. Simulation software is not always accurate, especially for complex materials or multi-phase injection processes. For this reason, it is advised that the authors include a discussion on software limitations, potential errors, and discrepancies between simulation and experimental results.
- Also I do have concerns about experimental validation. The validation approach is sound, but the study only compares a limited set of parameters. The comparison between simulation and experimental results shows some discrepancies in shrinkage %, yet there is no in-depth analysis of why these occur. Please provide statistical validation (e.g., RMSE, correlation coefficient) and discuss potential sources of error.
- Another issue is over-reliance on Autodesk Moldflow, because the study exclusively uses Autodesk Moldflow Synergy 2023, which limits its generalizability. The authors do not justify why this specific software was chosen over other options like Moldex3D or SIGMASOFT. It would be nice if authors can compare results from different simulation tools or provide justification for selecting Moldflow.
- In terms of the process window definition and scope, the process windows are defined primarily based on injection time, melt temperature, and mold temperature, but other critical parameters such as shear rate, cooling rate, and venting conditions are not considered. I am wondering if authors can expand the scope of process window definitions to include additional factors that impact part quality.
- The study is highly academic but does not fully discuss how the method can be practically implemented in real manufacturing environments. For improved industrial relevance, please provide case studies or discuss real-world applications from industry partners to show relevance.
- Most figures lack clarity, with axes labels and legends that are difficult to read. The process window graphs should include confidence intervals or error bars to indicate variability. It is highly advised that authors improve figure clarity, add error margins, and ensure proper labeling.
Improving the overall flow of English and contextual coherence would enhance readability and clarity
Reviewer 2 Report
Comments and Suggestions for Authors
Dear authors
Please note the following expressions:
UTILIZING SIMULATION SOFTWARE TO DEVELOP INJECTION MOLDING PROCESS WINDOWS
Abstract:
This paper studies injection molding. Because of the growth of injection molding industry, there is a need to use injection molding simulation softwares. this project focused on developing a method to construct process windows using injection molding simulation software with the process window split up into two stages that are filling and packing. In this article by usage of simulation softwares, they developed process windows for injection molding. They use 3 various CPV’s (controllable process variables) that if the Combination of CPVs is outside of the expected window ,causes defects in the manufactured part.
Reviewer Comments:
- They can use some techniques to decrease the number of CPV combinations that cause more accurate validation against low trials. And because of the increase in effective combination of CPV’s the extracted window is more accurate.
- One of these thenics is the Taguchi method that can make many effective choices and with wisdom trial instead of a high amount of trial and is one of the simplest and also wise methods.
- This article has a little grammar and so on defects that should be corrected but according to the manuscripts that I have judged up to now this manuscript has fewer of these mistakes.
- The tables are so clearly and appropriate in showing them and make the reader pleasant when he studies your manuscripts. In some manuscripts the tables are diffusing and makes readers uncomfortables and the readers scarcely continue up to the end of manuscripts.
- Text of the article is so clear and easy to understand and is continuously related to each other.
- Idea of the article is not so novel but is important and I think if they compare them with other types of manufacturing processes that are similar to it or bring CAE contours that reveal more characteristics of these CPVs and guide the reader to more information.
- The text referred to two of the most important things in this process such as filling and packing that must be shown and explained more of it in the article and showed some pictures of them.
- For more motivation of readers it is better to bring more pictures such as more pictures of ASTM machines and CPVs and other parameters of this manuscript.
- It is so important to introduce criterias for comprehension of damaged parts and bring some pictures of them. For example if they are experiencing fatigue you should introduce its related parameters or if the shape of the part is not like the desired part it should be inferred.
- If you used CAE methods you should say which software you used and which module or solver of them is used.
- You said in line 432 that “failure to eject a part can result in damage to the part along with the loss of cycle time” so you should infer the criteria of this damage and explain it in more details.
- Figure 1 should be more explained and The different parts of the photos should be pointed out and numbered, and each detail should be provided in each relevant issue with a more complete and complete explanation.
- You should infer the boundary conditions more accurately and divide them into mechanical and thermal ones and bring them into tables or pictures.
- Your title is so brief and clear that it gives the reader a good insight into the main purpose of the manuscript.
- Your abstract could be more brief instead of the presented one.
- Your conclusion is so clear and interesting that can help the reader to complete his understanding of total manuscript.
- It is really good to do some semi-analytical research in addition and do some optimization or add this application to your soft wares and windows.
- Figure 2 is so simple to understand, it is better to omit it or make it better.
Reviewer 3 Report
Comments and Suggestions for Authors
The paper presented here deals with the utilizing simulation software to develop injection molding process windows. The content of the paper is well developed. It may also bring practical application for practice. I include comments below that may improve the quality of the paper.
- The abstract should include numerical conclusions.
- The introduction section provides very little background from previous research. In addition, it is only a general description, so it is not adequate to engage the reader in my opinion. It needs to be reworked as it does not mention the shortcomings of the already existing studies. In the introductory section there are references to various references and what the researchers have looked at, but it is not clear what conclusions they have come to, their findings, and the results of their research. It would be good to add about 10 new references dealing with the issue to the introductory section.
- The novelty of the research needs to be clarified in the conclusion of the introduction section.
- Within the introductory section, I would recommend adding the following literature. Consider its use:
- Greškovič, F.; Varga, J.; Dulebová, Ľ. The utilize of gamma radiation on the examination of mechanical properties of polymeric materials. Metalurgija. 2012, 51(2).
- I miss an overall clear assessment of the overall results in the conclusion, in the form of an overall table or graph. Please add.
- In the final section, I am missing information regarding the focus for future research.
- Would it make practical sense in your opinion to apply your research to other types of materials?
I recommend the paper for publication after minor editing.
Round 2
Reviewer 2 Report
Comments and Suggestions for Authors
The manuscript is well-revised and can be published in present form.